# Integrating Individual Factors to Construct Recognition Models of Consumer Fraud Victimization

**DOI:** 10.3390/ijerph19010461

**Published:** 2022-01-01

**Authors:** Liuchang Xu, Jie Wang, Dayu Xu, Liang Xu

**Affiliations:** 1College of Mathematics and Computer Science, Zhejiang A&F University, Hangzhou 311300, China; xuliuchang@zafu.edu.cn (L.X.); xdysie@zafu.edu.cn (D.X.); 2Department of Psychology and Behavioral Sciences, Zhejiang University, Hangzhou 310058, China; wj_psy@zju.edu.cn

**Keywords:** consumer financial fraud, individual factors, machine learning, fraud exposure, fraud victimization

## Abstract

Consumer financial fraud has become a serious problem because it often causes victims to suffer economic, physical, mental, social, and legal harm. Identifying which individuals are more likely to be scammed may mitigate the threat posed by consumer financial fraud. Based on a two-stage conceptual framework, this study integrated various individual factors in a nationwide survey (36,202 participants) to construct fraud exposure recognition (FER) and fraud victimhood recognition (FVR) models by utilizing a machine learning method. The FER model performed well (f1 = 0.727), and model interpretation indicated that migration status, financial status, urbanicity, and age have good predictive effects on fraud exposure in the Chinese context, whereas the FVR model shows a low predictive effect (f1 = 0.565), reminding us to consider more psychological factors in future work. This research provides an important reference for the analysis of individual differences among people vulnerable to consumer fraud.

## 1. Introduction

The rapid development of mobile technology has facilitated social communication crossing geographical divides, but it has also increased the chances of fraudsters defrauding money [1]. In recent years, consumer financial fraud, defined as “crimes against consumers in which deceptive or false acts are committed for personal financial gain” [2,3], has attracted increasing attention from academia and government. Consumer financial fraud not only poses a threat to the consumer economy [4,5,6,7] but also causes victims to suffer physical, mental, social, and legal harm [8,9,10,11,12,13,14,15]. The great threat to public health makes anti-fraud an urgent issue.

Consumer financial fraud is essentially a process of deception used to defraud consumers of their money (and perhaps personal information) in a consumer context (e.g., when buying something). From the perspective of information transmission, Shadel and Pak indicated that fraudsters often portray themselves as a positive role model to convey fraudulent information, and some recipients of the information may believe it for specific reasons, leading to being deceived [16]. Grazioli used an information-processing model of deception detection to study the underlying reasons of consumers’ failure to detect intentional deception and found that the deceived usually rely on “trust” cues and heavily discount “assurance” cues [17]. After noting the importance of cue processing in the fraud process, Wright et al. proposed that individual factors also need to be considered [18]. The Process Model of Deception Detection improved by Wright et al. indicated that, in a fraud scenario, different individuals have different cue processing, evaluation, and decision-making methods [18]. Therefore, investigating who is more likely to be deceived has become a critical topic in fraud research.

Demographic factors, such as sex, age, education experience, and financial status, were the most frequently considered individual factors in previous fraud studies. For age, researchers found that older adults are less exposed to fraud because they are less likely to go online [18,19,20,21]. However, once older adults encounter fraud, they are more likely to become victims of fraud, that is, once they are targeted (exposed to fraud), they are more likely to be victimized. Older adults may face aging-related cognitive decline and retirement-related social isolation [4,22,23]. For financial status, Kerley and Copes found that people with incomes between USD 15,000 and 24,000 are more likely to be victims of fraud [19]. Sex and education experience may affect the individual’s probability of victimization in certain fraud scenarios. For example, although sex does not affect the victimization of fraud in general [19,24], males are more likely to be victims of investment fraud, lottery fraud, and advertising fraud [16,25]. For education experience, groups with low education levels are more likely to encounter loan and lottery fraud [26] but less likely to encounter investment fraud [27].

The above studies presented various pieces of evidence of the correlation between demographic factors and fraud, while psychologists are dedicated to finding the psychological factors that affect the victimization of fraud [28]. Previous psychological studies have shown that various psychological traits, such as personality, self control, aloneness, impulsion, cognitive ability, and emotional status, may affect the individual’s probability of being deceived [29,30,31,32,33,34,35,36]. For example, Gottfredson and Hirschi indicated that individuals with low self-control are more likely to be impulsive, focus on the present, and pursue immediate pleasure, so they are more likely to be involved in fraud [37]; van de Weijer and Leukfeldt found that individuals with low conscientiousness, low neuroticism, and high openness are more likely to be deceived [38]; inducing individual fear emotions can increase the individual’s trust in fraudulent information [39,40]; and Fischer et al. found that fraud victimization is related to high motivation, trust and excessive self-confidence [41]. In addition, according to the depth interviews with fraud victims, Button et al. summarized a series of reasons for the victims of online fraud, including embarrassing fraud, visceral appeals, pressure, and other psychological factors [42]. These psychological studies have given us a deeper understanding of the individual factors that affect the victimization of fraud.

Whether studying demographic or psychological factors, these studies help us understand why some individuals appear to be more susceptible to being deceived [43]. However, most of the studies have focused on a single or small number of factors and rarely considered different types of individual factors together. With the development of machine learning (ML) technology, it is possible to compare the effects of different features from the perspective of computational modeling [44]. Thus, can we use ML technology to compare the importance of individual factors in fraud scenarios? Can we build a predictive model to identify which individuals are more susceptible to fraud? These are the questions that this research intends to explore.

In fact, in different stages of fraud, the role of individual factors may also be different. As mentioned before, older adults are less exposed to fraud but more likely to become victims of fraud once exposed [16,22]. This reminds us that fraud has two important stages: fraud exposure and fraud victimization. Fan and Yu recently developed a two-stage conceptual framework to investigate age-related differences in fraud exposure and fraud victimization [4]. The first stage of fraud exposure was defined as whether consumers experienced fraud regardless of whether they were victimized, and the second stage was whether consumers became victims (lost money) of fraud that they were exposed to [4]. The above work found that some factors (e.g., sex, assets, and region) have a significant impact on fraud exposure but have no influence on whether they are victimized. In fact, previous research on fraud mechanisms has mainly focused on investigating the second stage of fraud, that is, why some people are more easily deceived [31,43]; however, there is less research on the first stage. Therefore, to better investigate fraud mechanisms, this study built two predictive models of fraud exposure and fraud victimization.

In summary, the present study uses individual factors as inputs to construct fraud exposure recognition (FER) and fraud victimhood recognition (FVR) models. From a practical perspective, we built a predictive model to automatically identify which individuals are more likely to be exposed to fraud and which individuals are more susceptible to fraud. Identifying individuals susceptible to fraud in advance may help us reduce the number of victims. From a theoretical perspective, we used a computational modeling method to compare the impact of various individual factors on fraud exposure and fraud victimhood. The results of factor comparison can provide an important reference for subsequent research on individual differences in fraud.

## 2. Materials and Methods

### 2.1. Data and Participants

The present work used the China Household Finance Survey (CHFS) microdata, a public-use database collected by the Survey and Research Center for China Household Finance [45], to achieve our objectives. The CHFS is a nationwide biennial survey of Chinese household finances containing information about household demographics, geographic location, assets and liabilities, income and expenditure, employment, consumer fraud, and so forth [4,45]. The 2015 CHFS database (Accessible at https://chfser.swufe.edu.cn/datas/Products/Datas/DataList, accessed date 10 December 2020) [45], including 37,289 households in 351 counties of 29 Chinese provinces, was applied in this study. After removing samples with severe missing data, the data of 36,202 households were considered in subsequent modeling and analysis. In addition, since the person who is the most knowledgeable about their household finance was interviewed in CHFS [4], he or she was regarded as the household reference person here. The average age of the final 36,202 participants (47.29% females) was 52.83 years (SD = 14.95).

### 2.2. Measurements and Feature Processing

This section describes how the variables were measured in the 2015 CHFS and how they were processed for subsequent modeling. The main features, including fraud exposure, fraud victimhood, demographic features, and financial-related features, are introduced in detail. However, considering the length of the article, the complete feature list (a total of 150 features) is presented in Appendix A.

#### 2.2.1. Fraud Exposure and Victimhood

Consumer fraud exposure was investigated before fraud victimhood. First, the survey asked “whether the household encountered the following consumer fraud over the past year”, including telephone fraud, SMS fraud, social application (such as QQ and WeChat) fraud, phishing, fraud from acquaintances, and other methods. Multiple options were allowed for this question. If the respondent encountered at least one of the fraud methods, the household was considered exposed to consumer fraud, and then he or she was asked “whether the household has suffered monetary losses as a result of fraud”. If the respondent answered “yes”, the household was regarded as a consumer fraud victim. Consumer fraud exposure and victimhood were treated as ground truth in our predictive models. As binary features, they were labeled as 1 (exposed to consumer fraud; fraud victim) or 0 (not exposed to consumer fraud; fraud survivor).

#### 2.2.2. Demographic Features

Demographic information included age, sex (male of female), education (from 1 no schooling at all to 9 doctorate degree), marital status, employment (having a job or not), self-evaluation of physical condition, migration status (having which type of registered residence), region (eastern, western, or central China), resident type (rural or urban), financial knowledge, and political status (whether a Chinese Communist Party member). As model inputs, all single-choice multi-categorical features were labeled as different numbers from 1 to *n* (*n* is the number of options), all multiple selection features were transformed into dummy variables, and all continuous inputs were scaled to a value between 0 and 1. The above feature-processing methods were also used to process other input features.

#### 2.2.3. Financial-Related Features

Defrauding money is the main purpose of customer financial fraud [3], so the individual’s financial-related features are the key considerations of this research. The 2015 CHFS collected information about assets, liabilities, income, and expenditures, and each feature had multiple molecular dimensions. For instance, income information included not only the total income but also sources of income, such as lottery winnings, the sale of a house, sale of intellectual property, and so forth; expenditures included expenditures on different items, such as food, transportation, communication, luxury, education, and so forth. Most of the molecular dimensions were transformed into independent features as model inputs.

#### 2.2.4. Other Features

In addition to the above main features, other potentially relevant features, such as risk appetite and subjective well-being, were also treated as model inputs. For example, participants’ risk appetite was considered an input in this study because risk appetite may influence an individual’s financial behavior [46,47]. The risk appetite was investigated by two questions (e.g., “Which of the choice below do you want to invest most if you have adequate money?” from 1 “project with high-risk and high-return” to 5 “unwilling to carry any risk”), and the answer to each question was used as an input feature. Subjective well-being was measured by asking “how happy does the respondent feel”, and the respondent was asked to respond on a scale from 1 “extremely happy” to 5 “extremely unhappy”. In sum, the name and description of all features are presented in Appendix A.

### 2.3. Model Construction and Evaluation

The present work constructed two recognition models of fraud exposure and fraud victimhood. Both fraud exposure and victimhood were binary features, so we formulated fraud exposure recognition and fraud victimhood recognition as classification problems. We used fraud exposure and fraud victimhood as ground truth, used the other features introduced above as inputs, and applied a machine learning algorithm to construct the predictive models.

For the machine learning algorithm, this work applied the random forest classification (RFC) algorithm. As RFC has shown good performance in classification tasks [48,49], we can easily interpret the constructed RFC models by calculating the feature importance [50,51,52,53,54]. The predictive effect of each model was evaluated by the tenfold cross-validation technique. The tenfold cross-validation technique uses 90% of the data as training data to train the models and the remaining instances as testing data, and this procedure is repeated ten times [50]. Finally, the prediction accuracy of each classifier was measured using precision, recall, and F1 values as follows [55]:Precision = TP/(TP + FP)(1)
Recall = TP/(TP + FN)(2)
F1 = 2 × Precision × Recall/(Precision + Recall)(3)
where TP (true positive) is the number of positive samples predicted by the classifier as positive; FP (false positive) is the number of negative samples predicted by the classifier as positive; and FN (false negative) is the number of positive samples predicted by the classifier as negative.

## 3. Results

### 3.1. Basic Statistics

As the first step of data exploration, several basic statistics of the dataset were examined. As shown in Table 1, among all the participants, 58.62% (*n* = 21,221) reported being exposed to fraud, but among those who were scammed, only 6.05% (*n* = 1284) became fraud victims (losing money). For age, we observed that younger people were more likely to be exposed to fraud (F(1, 36,201) = 395.051, *p* < 0.001, *η^2^* = 0.011). However, once exposed to fraud, the mean age of the victims was older than that of the survivors (F(1, 21,220) = 32.002, *p* < 0.001, *η^2^* = 0.002). The financial status of an individual affects the probability of an individual encountering fraud. Compared with individuals not exposed to fraud, individuals exposed to fraud have more assets (F(1, 36,201) = 899.874, *p* < 0.001, *η^2^* = 0.024), debts (F(1, 36,201) = 89.349, *p* < 0.001, *η^2^* = 0.002), income (F(1, 36,201) = 291.647, *p* < 0.001, *η^2^* = 0.008), and consumption (F(1, 36,201) = 666.952, *p* < 0.001, *η^2^* = 0.018). Once exposed to fraud, the financial status does not affect whether individuals are victimized. These results describe the linear relationship between several individual factors and fraud. The individual factors of finer granularity and the nonlinear relationship between variables were analyzed through subsequent computational modeling analysis.

### 3.2. Model Prediction Results

This study used fraud exposure and fraud victimhood as ground truth, used various individual factors as inputs, and applied the RFC algorithm to construct predictive models. In addition, only 6.05% of the participants were victimized after being exposed to fraud. Therefore, for the FVR model, random undersampling was used to balance the number of positive and negative samples [28]. For both the FER and FVR models, a grid parameter search was applied to select the best parameters, and the best parameters for each model are presented in Table 2.

The performance of the FER model is shown in Figure 1. We observed that the FER model achieved a mean precision value of 0.733, a mean recall value of 0.721, and a mean f1 value of 0.727 (see Figure 1a). The receiver operating characteristic (ROC) curve of our FER model is shown in Figure 1b, and we observed that the mean area under the curve (AUC) was 0.675. The FVR model reached a mean precision value of 0.580, a mean recall value of 0.553, and a mean f1 value of 0.565 (see Figure 2a). The ROC curve of our FVR model is shown in Figure 2b, and the mean AUC was 0.577. In general, the performance of the FER model was better than that of the FVR model (f1: t(9) = 10.405, *p* < 0.001, *d* = 3.290).

### 3.3. Model Interpretability

We then explained the constructed models by examining the information gain of features (calculating feature importance) [51]. The feature importance of the FER and FVR models is presented in Figure 3 and Figure 4, respectively. Since tenfold cross-validation technology was used to evaluate our models, the feature importance was different when predicting different test sets. Therefore, the distribution of feature importance was arranged in descending order of the mean value, and only the top 20 features were included for visibility. We observed that for the FER model, the most important feature is a registered residence, accounting for 6.98% of the model. Total assets are the second most important feature (4.93%), followed by funds (4.07%), total consumption (3.66%), rural areas (3.50%), total income (3.05%), stock accounts (2.75%), and so on. For the FVR model, the total income is the most important feature (7.34%), followed by age (6.33%), clothing expenses (4.82%), total assets (4.46%), cash (4.39%), and total consumption (3.67%). The above results compare the potential impact of different features on fraud exposure and fraud victimization, and the complete importance scores are presented in Appendix A.

## 4. Discussion

The present work used a computational modeling method to examine the impacts of various individual factors on consumer financial fraud. Based on the two-stage conceptual framework of fraud [4], we used a nationwide dataset (CHFS) [45] to construct the fraud exposure recognition and fraud victimhood recognition models. The importance of the model features indicates which individuals are more likely to be exposed to fraud and which individuals are more likely to be victimized by fraud. The model we build also has practical value because early identification of vulnerable individuals may be able to reduce the harm caused by fraud by intervening early. Our models may also be useful in practices in finance area, such as in the mobile banking applications.

The basic statistics of the CHFS dataset were first conducted to explore the linear relationship among several individual factors, fraud exposure, and fraud victimhood. The results show that people exposed to fraud were younger than those not exposed, whereas the victims were older than the survivors once exposed to fraud. This finding supports the previous opinion that older adults are less exposed to fraud but more likely to become victims of fraud [16,22]. We also found that individuals with more assets, debts, incomes, and consumption were more likely to be exposed to fraud. This result reflects that more economic activity or possession of more assets increases the likelihood of being the target of consumer financial fraud because defrauding money is the main purpose of customer financial fraud [3]. However, similar to previous findings [31], financial factors do not have any significant relationship with whether the individual will become a victim. More factors were then explored in subsequent modeling.

The FER model we constructed has a predictive effect, reaching a mean f1 value of 0.727. Identifying individuals who are more vulnerable to fraud may be able to directly prevent individuals from being exposed to fraud. Therefore, we believe that using the current prediction model may indirectly mitigate the threat posed by consumer financial fraud. The FER model was then interpreted by calculating feature importance. The results show that the migration status (i.e., which type of registered residence) is the most important factor in fraud exposure. Although the socioeconomic and health disadvantages of Chinese migrants have been reviewed in previous works [56,57], the fact that being migrants put consumers at a substantially higher risk of being targeted by perpetrators was only recently discovered [4]. Our results further lay the foundation for the decisive impact of immigration on fraud exposure. Similar to the basic statistical results, many financial-related features (e.g., assets, total consumption, funds, income, food expenses, and living expenses) play an important role in predicting fraud exposure. This result once again supports the opinion that defrauding money is the main purpose of customer financial fraud [3]. We also observed that, in the Chinese context, urbanicity (whether living in rural areas) and age have important influences on whether fraud will be encountered. Further statistical analysis shows that living in rural areas and being older decreased the chance of being a fraud target, similar to previous survey results [19,20]. Finally, although other factors, such as the opinion of banking services (accounting for 1.98% of the model), attitudes toward online financial products (1.26%), and risk appetite (0.79%), also have predictive effects, their effects are much smaller than the features mentioned above.

The FVR model shows a low predictive effect on fraud victimhood, only reaching a mean f1 value of 0.565. Model interpretation shows that a large number of financial-related features, such as income, clothing expenses, consumption, and debt, played important roles in the FVR model, although statistical tests were not significant. Both the model recognition results and the feature importance results remind us that the individual factors considered in this study are not sufficient to effectively predict fraud victimization. Predicting whether someone will be victimized may require more consideration of the influence of individual cognition and psychological factors [28,58]. For FER, its essence is to investigate the criteria for fraudsters to choose fraud targets. Individuals’ demographic, economic and geographic factors considered in this study are also decisive factors for fraudsters. However, once exposed to fraud (i.e., for FVR), an individual’s psychological factors may determine whether he or she will be a victim of fraud. Previous works have shown that victims often fail to recognize deception cues due to psychological factors, such as personality, low self-control, and impulsivity [29,32,34,58]. Vishwanath et al. found that individuals holding the psychological characteristics associated with victimhood are more likely to use heuristics to make quick (usually erroneous) decisions [35]. Unfortunately, few psychological factors were investigated in the CHFS survey, and the considered factors, including risk appetite and well-being, have little effect on fraud victimhood. Therefore, measuring and integrating various psychological factors to construct FVR models is an important direction for future research.

In addition, the different types of scams are not distinguished in this study. In fact, the victims of different scams may have different characteristics. For example, according to data from Tencent 110, men are more likely to be scammed in pornography and dating scams, while women are less likely to be scammed in these two types of scams [59]. At the same time, the incidence of different types of scams may vary in different countries. In China, transaction scams and loan scams are more common, followed by identity impersonation scams, financial management scams, and online dating scams [60]. For future research in the Chinese context, separate modeling for more high-incidence scams can be considered to improve the accuracy of recognition models.

From a practical perspective, our recognition models can be applied to combat fraud. The FER model can be used to identify who are more likely to be the targets of fraud and the FVR model can be used to predict who are more likely to be defrauded and suffer losses (although our results showed that this model need more psychological factors). For government departments, after identifying susceptible groups, they can carry out more anti-fraud training and publicity for these people to improve their anti-fraud consciousness and avoid victimization. For banks, they can identify vulnerable users by using our recognition models, so that certain security measures can be taken for these users (such as blocking possible fraudulent transactions). In addition, banks can also consider the recognition results of our models in the fraudulent financial transaction detection model. Referring to our recognition models, the accuracy of fraud detection may be improved. Notably, similar to fraud detection, our recognition models also rely on consumers’ demographic data and financial data, so privacy intrusiveness should be considered when collecting information [61].

## 5. Conclusions

In conclusion, the present study integrated a large number of individual factors to predict customer fraud victimization. Based on the two-stage conceptual framework, we constructed FER and FVR models, respectively. The FER model performed well, and model interpretation indicated that migration status, financial status, urbanity, and age have good predictive effects on fraud exposure in the Chinese context, whereas the FVR model shows a low predictive effect, reminding us to consider more psychological factors in future work.

## Figures and Tables

**Figure 1 ijerph-19-00461-f001:**
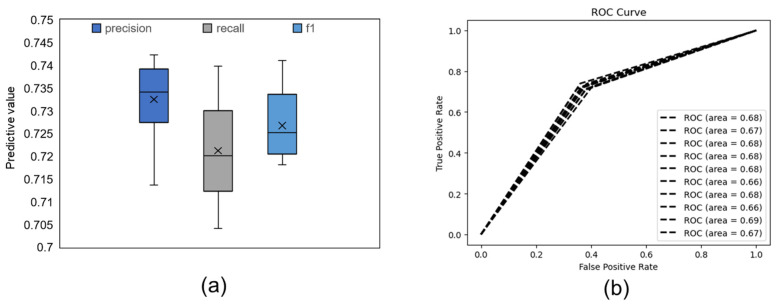
Predictive results of the FER model. (**a**) Each box shows the distribution of the predictive results of ten test sets. The black line in the middle of each box indicates the median; “×” indicates the mean value. (**b**) ROC curve of ten test sets.

**Figure 2 ijerph-19-00461-f002:**
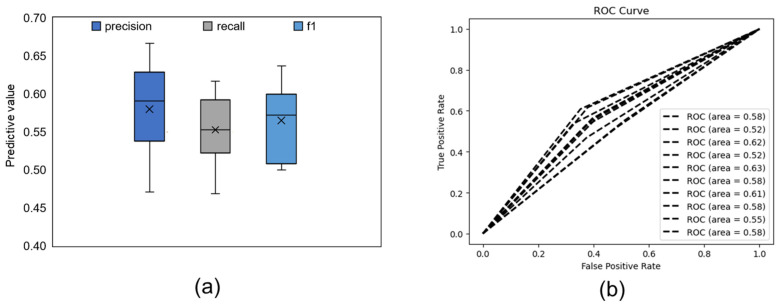
Predictive results of the FVR model. (**a**) Each box shows the distribution of the predictive results of ten test sets. The black line in the middle of each box indicates the median; “×” indicates the mean value. (**b**) ROC curve of ten test sets.

**Figure 3 ijerph-19-00461-f003:**
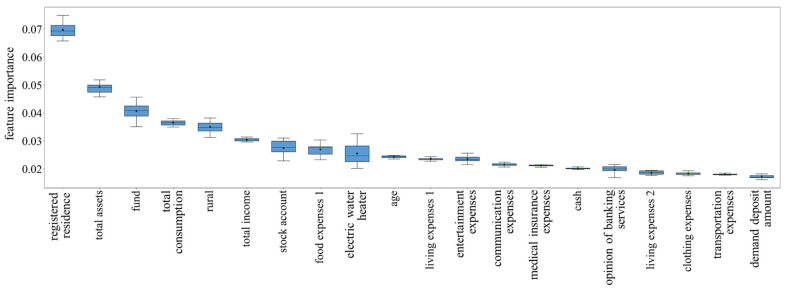
Distribution of feature importance for FER model. Arranged in descending order of the mean value, the top 20 features were included for visibility, and the trend of the remaining features was approximately the same. Error bars indicate the standard deviations, the black dots indicate the mean values, and the black dotted lines indicate the median values.

**Figure 4 ijerph-19-00461-f004:**
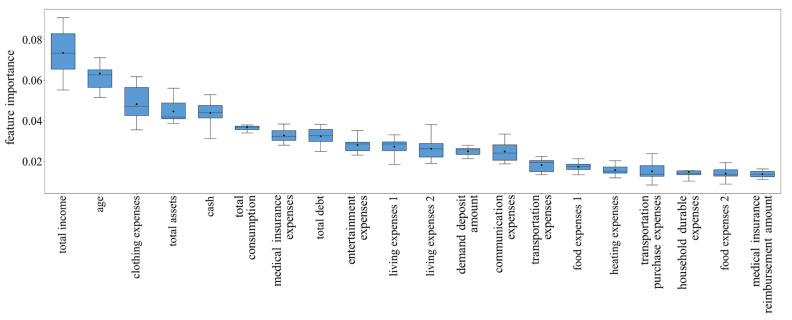
Distribution of feature importance for FVR model. Arranged in descending order of the mean value, the top 20 features were included for visibility, and the trend of the remaining features was approximately the same. Error bars indicate the standard deviations, the black dots indicate the mean values, and the black dotted lines indicate the median values.

**Table 1 ijerph-19-00461-t001:** The basic statistics for the dataset.

Individual Factors		Fraud Exposure (*n* = 36,202)		Fraud Victimhood (*n* = 21,221)	
		Exposed (*n* = 21,221)	Not Exposed (*n* = 14,981)	Fraud Victim (*n* = 1284)	Fraud Survivor (*n* = 19,937)
Age (M ± SD)		51.52 ± 14.88	54.68 ± 14.85	53.80 ± 16.44	51.37 ± 14.77
Sex (%)	Male (Female)	52.20 (47.80)	53.44 (46.56)	52.80 (47.20)	52.15 (47.85)
Asset (¥)	M (SD)	1.15×10^6^ (2.03×10^6^)	5.80×10^5^ (1.34×10^6^)	1.04×10^6^ (1.83×10^6^)	1.16×10^6^ (2.05×10^6^)
Debt (¥)	M (SD)	9.17×10^4^ (2.16×10^5^)	5.70×10^4^ (1.47×10^5^)	8.18×10^4^ (2.15×10^5^)	9.23×10^4^ (2.16×10^5^)
Income (¥/year)	M (SD)	5.67×10^4^ (2.49×10^5^)	3.42×10^4^ (1.81×10^5^)	6.39×10^4^ (2.17×10^5^)	5.62×10^4^ (2.50×10^5^)
Consumption (¥/year)	M (SD)	6.57×10^4^ (7.51×10^4^)	4.62×10^4^ (6.36×10^4^)	6.54×10^4^ (7.25×10^4^)	6.57×10^4^ (7.53×10^4^)

*n* indicates the number of participants, M indicates the mean values, and SD indicates standard deviation.

**Table 2 ijerph-19-00461-t002:** The best parameters for RFC models.

Parameters	Models	
	FER	FVR
n_estimators	127	47
max_depth	20	10
min_samples_leaf	5	10
min_samples_split	35	45
max_features	0.3	0.9

FER indicates fraud exposure recognition and FVR indicates fraud victimhood recognition.

## Data Availability

The raw data are available at https://chfser.swufe.edu.cn/datas/Products/Datas/DataList (accessed date 10 December 2020).

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
