# Peer review of "Integrating Individual Factors to Construct Recognition Models of Consumer Fraud Victimization"

_ijerph, 2022, doi:10.3390/ijerph19010461_

Round 1
Reviewer 1 Report
The article Integrating Individual Factors to Construct Recognition Models 2 of Consumer Fraud Victimization presents a significant problem of consumer financial fraud. In the papier the auteurs uses individual factors as inputs to construct fraud exposure recognition (FER) and fraud victimhood recognition (FVR) models. They have built and test a predictive model to automatically identify which individuals are more likely to be exposed to fraud and which individuals are more susceptible to fraud. In my opinion presented model may by very useful in practices in finance area.
In my opinion, it might be very interesting to enhance the presented model with technical aspect such as understanding the technology (threats) that are used by customers in finance area f.e. banks mobile applications.
Author Response
Thanks for your helpful comments about our manuscript! We agree with you that our models may be used in banking applications. We have added a brief mention about our models’ practical values in the manuscript as follows (page 8, lines 274): “Our models may also be useful in practices in finance area, such as in the mobile banking applications.”
Reviewer 2 Report
This is an interesting paper that seeks to develop a model for the prediction of consumer fraud. With some further amendments and additions it will be a suitable paper to publish.
On page 1 "Consumer financial fraud is essentially a process in which consumers are persuaded by fraudsters." Is used as the working definition. This is weak and doenst offer much sophistication. Surely a process of deception used to trick consumers of their money (perhaps personal information) in a consumer context ie, when buying something. I suggest the authors look at this as the definition sets up the whole article and at present this is 'sand foundations'.
The literature review on falling victim was a bit light on key literature. The authors should consult the following: Fischer, P., Lea, S. E., & Evans, K. M. (2013). Why do individuals respond to fraudulent scam communications and lose money? The psychological determinants of scam compliance. Journal of Applied Social Psychology, 43(10), 2060-2072; and Button, M., Nicholls, C. M., Kerr, J., & Owen, R. (2014). Online frauds: Learning from victims why they fall for these scams. Australian & New Zealand journal of criminology, 47(3), 391-408.
The impact section was also dated and light. See Button, M., Lewis, C., & Tapley, J. (2014). Not a victimless crime: The impact of fraud on individual victims and their families. Security Journal, 27(1), 36-54; Cross, C. (2016). ‘They’re very lonely’: Understanding the fraud victimisation of seniors. International Journal for Crime, Justice and Social Democracy, 5(4), 60; and far East perspectives too, such as Kadoya, Y., Khan, M. S. R., Narumoto, J., & Watanabe, S. (2021). Who is next? A study on victims of financial fraud in Japan. Frontiers in psychology, 12, 2352 and also consider some of the work on China Lee, C. S. (2021). How Online Fraud Victims are Targeted in China: A Crime Script Analysis of Baidu Tieba C2C Fraud. Crime & Delinquency, 00111287211029862.
I also wasnt sure about this statement "However, once older adults encounter fraud, they are more likely to become victims of fraud." Its repeated again later. What do the authors mean, is it once they go online or if they are targted? More explanation of this is required. If it is the former it implies routine activity theory too, which hasnt been noted and the authors should look at Pratt, T. C., Holtfreter, K., & Reisig, M. D. (2010). Routine online activity and internet fraud targeting: Extending the generality of routine activity theory. Journal of Research in Crime and Delinquency, 47(3), 267-296.
I think readers will also be interested to have some brief context on the nature of consumer fraud in China. Common types of consumer fraud, extent etc.
If these issues can be addressed the paper deserves publication.
Author Response
Thanks for your helpful comments about our manuscript. Following the insightful comments and helpful suggestions, we have revised the manuscript, and the revised parts have been highlighted as yellow. Our point-by-point responses to your comments are as follows.
- On page 1 "Consumer financial fraud is essentially a process in which consumers are persuaded by fraudsters." Is used as the working definition. This is weak and doenst offer much sophistication. Surely a process of deception used to trick consumers of their money (perhaps personal information) in a consumer context ie, when buying something. I suggest the authors look at this as the definition sets up the whole article and at present this is 'sand foundations'.
Response: Thanks for this insightful comment! We have revised the definition of consumer financial fraud in the manuscript as follows (page 1, line 33): “Consumer financial fraud is essentially a process of deception, used to defraud consumers of their money (and perhaps personal information) in a consumer context (e.g., when buying something).”
- The literature review on falling victim was a bit light on key literature. The authors should consult the following: Fischer, P., Lea, S. E., & Evans, K. M. (2013). Why do individuals respond to fraudulent scam communications and lose money? The psychological determinants of scam compliance. Journal of Applied Social Psychology, 43(10), 2060-2072; and Button, M., Nicholls, C. M., Kerr, J., & Owen, R. (2014). Online frauds: Learning from victims why they fall for these scams. Australian & New Zealand journal of criminology, 47(3), 391-408.
Response: Thanks for the great literature! We have supplemented them in the manuscript as follows (page 2, lines 70): “Fischer et al. found that fraud victimization is related to high motivation, trust and excessive self-confidence [53]. In addition, according to the depth interviews with fraud victims, Button et al. summarized a series of reasons for the victims of online fraud, including embarrassing fraud, visceral appeals, pressure and other psychological factors [54].”
- The impact section was also dated and light. See Button, M., Lewis, C., & Tapley, J. (2014). Not a victimless crime: The impact of fraud on individual victims and their families. Security Journal, 27(1), 36-54; Cross, C. (2016). ‘They’re very lonely’: Understanding the fraud victimisation of seniors. International Journal for Crime, Justice and Social Democracy, 5(4), 60; and far East perspectives too, such as Kadoya, Y., Khan, M. S. R., Narumoto, J., & Watanabe, S. (2021). Who is next? A study on victims of financial fraud in Japan. Frontiers in psychology, 12, 2352 and also consider some of the work on China Lee, C. S. (2021). How Online Fraud Victims are Targeted in China: A Crime Script Analysis of Baidu Tieba C2C Fraud. Crime & Delinquency, 00111287211029862.
Response: The suggested literature is helpful! We have read and quoted them in the manuscript as follows (page 1, lines 29): “Consumer financial fraud not only poses a threat to the consumer economy [10-12, 58] but also causes victims to suffer physical, mental, social, and legal harm [13-17,55-57].”
- I also wasn’t sure about this statement "However, once older adults encounter fraud, they are more likely to become victims of fraud." Its repeated again later. What do the authors mean, is it once they go online or if they are targted? More explanation of this is required. If it is the former it implies routine activity theory too, which hasn’t been noted and the authors should look at Pratt, T. C., Holtfreter, K., & Reisig, M. D. (2010). Routine online activity and internet fraud targeting: Extending the generality of routine activity theory. Journal of Research in Crime and Delinquency, 47(3), 267-296.
Response: We are sorry for not clearly expressing the statement. The meaning of this statement refers to the latter you mentioned, that is, once the elderly become targets of fraud (exposed to fraud), they are more likely to become victims. This corresponds to the second stage of the two-stage conceptual framework of fraud: the victimhood stage after exposure to fraud. We have added an explanation of this statement in the manuscript as follows (page 2, lines 50-52): “However, once older adults encounter fraud, they are more likely to become victims of fraud, that is, once they are targeted (exposed to fraud), they are more likely to be victimized.” And the former you mentioned (once they go online) corresponds to this statement “For age, researchers found that older adults are less exposed to fraud because they are less likely to go online.” As you said, this statement does imply routine activity theory, so we have also added a reference to the article by Pratt et al. in the manuscript (page 2, lines 48-50) as follows: “For age, researchers found that older adults are less exposed to fraud because they are less likely to go online [5,25,26, 59].”
- I think readers will also be interested to have some brief context on the nature of consumer fraud in China. Common types of consumer fraud, extent etc.
Response: Thanks for this helpful suggestion! We have added some description about the nature of consumer fraud in China in the discussion section as follows (page 9, lines 329-337): “In addition, the different types of scams are not distinguished in this study. In fact, the victims of different scams may have different characteristics. For example, according to data from Tencent 110, men are more likely to be scammed in pornography and dating scams, while women are less likely to be scammed in these two types of scams [60]. At the same time, the incidence of different types of scams may vary in different countries. In China, transaction scam and loan scam are more common, followed by identity impersonation scam, financial management scam, and online dating scam [61]. For future research in the Chinese context, separate modeling for more high-incidence scams can be considered to improve the accuracy of recognition models.”
Reviewer 3 Report
The section of “concrete” proposals to be implemented for combating fraud (specific to the respective area - based on Recognition Models) is missing. Should be completed.
Also, if it is considered to be in line with the topic of the article, can be quoted “Găbudeanu, Larisa, Iulia Brici, CodruÈ›a Mare, Ioan Cosmin Mihai, and Mircea Constantin Șcheau. 2021. Privacy Intrusiveness in Financial-Banking Fraud Detection. Risks 9: 104. https://doi.org/10.3390/risks9060104”
Author Response
Thanks for your helpful comments about our manuscript. Following the insightful comments and helpful suggestions, we have revised the manuscript, and the revised parts have been highlighted as yellow. Our point-by-point responses to your comments are as follows.
- The section of “concrete” proposals to be implemented for combating fraud (specific to the respective area - based on Recognition Models) is missing. Should be completed.
Response: thanks for your useful suggestion! In the application of our models, more specific implementation proposals are indeed needed. We have added this part at the end of the discussion in the manuscript as follows (page 9, lines 338-348): “From a practical perspective, our recognition models can be applied to combat fraud. The FER model can be used to identify who are more likely to be the targets of fraud and the FVR model can be used to predict who are more likely to be defrauded and suffer losses (although our results showed that this model need more psychological factors). For government departments, after identifying susceptible groups, they can carry out more anti-fraud training and publicity for these people to improve their anti-fraud consciousness and avoid victimization. For banks, they can identify vulnerable users by using our recognition models, so that certain security measures can be taken for these users (such as blocking possible fraudulent transactions). In addition, banks can also consider the recognition results of our models in the fraudulent financial transaction detection model. Referring to our model our recognition models, the accuracy of fraud detection may be improved.”
- Also, if it is considered to be in line with the topic of the article, can be quoted “Găbudeanu, Larisa, Iulia Brici, CodruÈ›a Mare, Ioan Cosmin Mihai, and Mircea Constantin Șcheau. 2021. Privacy Intrusiveness in Financial-Banking Fraud Detection. Risks 9: 104.https://doi.org/10.3390/risks9060104”
Response: Thanks for the great articles! The section of “concrete” proposals to be implemented for combating fraud is really related to our work, so we have quoted this paper in our manuscript as follows (page 9, lines 348-351): “Notably, similar to fraud detection, our recognition models also rely on consumers’ demographic data and financial data, so privacy intrusiveness should be considered when collecting information [62].”